# Review of Polymetallic Mineralization in the Sila and Serre Massifs (Calabria, Southern Italy)

Rosa Anna Fregola [1,*], Antonio Ciccolella [1], Vincenzo Festa [1], Giovanni Ruggieri [2], Emanuela Schingaro [1], Fabrizio Tursi [3] and Gennaro Ventruti [1]

1   Dipartimento di Scienze della Terra e Geoambientali, Università degli Studi di Bari Aldo Moro, Via Orabona 4, 70125 Bari, Italy

2   Istituto di Geoscienze e Georisorse (IGG), Consiglio Nazionale delle Ricerche (CNR), Via La Pira 4, 50121 Firenze, Italy

3   Dipartimento di Scienze della Terra, Università degli Studi di Torino, Via Valperga Caluso 35, 10125 Torino, Italy

\*   Correspondence: rosaanna.fregola@uniba.it

**Abstract:** We provide an updated overview of the known mineral deposits from the Sila and Serre Massifs in Calabria, contributing to setting their genesis within a complex geologic history, starting from the late-Carboniferous. We summarize the mineralization reported in the literature, with a critical review of the host tectonic units, by taking into account the upgrades in the knowledge of these areas. We also set them in updated geological maps and in stratigraphic columns, highlighting the crustal levels to which they pertain. Despite the geologic and minerogenetic similarities potentially existing with late- to post-Variscan mineral deposits from other regions (e.g., Sardinia and French Central Massif), the scientific literature on the Calabria mineralization is out-of-date and not exhaustive. Moreover, these ore deposits were likely considered not economically attractive enough to stimulate new scientific studies. However, in our opinion, such studies are needed to resolve the main open questions, which rely on deciphering the origin and age of mineralization. Finally, research for critical elements hosted by the Sila and Serre mineralization (e.g., In, Ge and Ga in sphalerites) is a possible interesting new perspective.

**Keywords:** mineralization; Sila; Serre; ore deposits; polymetallic sulphides; ore genesis

## 1. Introduction

The study of mineralization from metal-carrying fluids during orogenic events is essential to assess the budget and availability of elements in their life cycle, from the deep Earth's lithosphere to the atmosphere, biosphere and hydrosphere (e.g., [1]). Hence, it is important to outline the different types of mineralization developed during poly-orogenic events into the tectonic, metamorphic, magmatic and sedimentary evolution of a geologic area, in order to relate deep Earth volatiles and metal factories into the different geodynamic settings.

Our paper aims to provide an updated and extensive overview of the scientific literature on mineral deposits in the Sila and Serre Massifs from Calabria (southern Italy). Calabria has been a crucial district for raw metals exploitation starting from the Bronze and Iron ages. Recent historical and archaeological literature on the Calabria ores and mines (e.g., [2,3]) is wider than the scientific one. Particularly, focus on the iron mining industry, once active in the Stilaro valley of the Serre area, is found in the Italian archeo-minerary literature [4–6].

The mineralizations in Calabria have been described throughout time, for both industrial exploitation and geological interest. In particular, the first reports came between the first half of the 19th century and the 1940's [7–30]. In order to ascertain the mining potential of Calabria, and to determine the relationships between host rocks and mineralization,

a field geological survey was started in 1946 by "Centro Ricerche Geominerarie" of the Italian Institute for Industrial Reconstruction [31], and by "Centro Studi Silani" of the Italian C.N.R. [32]. At the beginning, the study was restricted to a few localities of the Sila [33–37] and Serre Massifs [38]. After that, some prospecting campaigns were performed by the Italian "Comitato Nazionale per l'Energia Nucleare" [39–44]. In 1974, the Italian Government recommended a regional geochemical survey as a key tool for a mineral potential assessment program in Calabria. This program was performed by a mining research company (RIMIN Spa), by the Italian "Ministero dell'Industria, Commercio e Artigianato" (MICA), and by "Ente Nazionale Idrocarburi" (ENI); geochemical analyses were conducted on thousands of samples collected from different localities throughout Calabria [45–50]. The analyses revealed positive anomalies of Pb, Zn, Mo, As, W, Ba, Sn, Ag and Au, for the Rossano-Longobucco-Savelli areas (Sila); As, Be, Ba, Pb and Zn, to the north of Catanzaro (Sila); Hg and As, to the south of the Savuto river (Sila); and Pb, As, Zn and Mo, in the western part of the Stilo area (Serre). Results from samples collected during the geochemical surveys of the 1970′s and 1980′s were subsequently used to compile a geochemical baseline map illustrating the regional variability of twelve selected elements: Cu, Pb, Zn, Hg, Fe, Mn, Ba, As, W, Mo, Sn and Be [51].

Early efforts to survey the known scientific literature on mineralization from the Sila and Serre Massifs were made, trying to infer some hypotheses on the ore genesis. These hypotheses were generally based on lithostratigraphic considerations rather than on analytical results (e.g., [52–56]).

We provide a critical review of the literature on the polymetallic mineralizations in the Sila and Serre Massifs, with emphasis on the still-open issue of the ore genesis, and we build up a dataset that paves the way to future research on the subject. This dataset reviews the lithostratigraphic approach of the most relevant background literature on this topic (e.g., [53–55,57]) by trying to solve the emerging inconsistencies with respect to the present geologic knowledge on the poly-orogenic evolution of the Calabria's crystalline basement. The main areas of mining interest in the middle-eastern sectors of Calabria that will be considered (Figure 1) are: (i) Longobucco-Corigliano-Rossano, San Giovanni in Fiore-Savelli-Cerenzia, Catanzaro-Tiriolo-Gimigliano (in the Sila territory); and (ii) Stilo-Pazzano-Bivongi, San Todaro-Calatria-Caulonia (in the Serre area). A comprehensive account on ore geology will be presented following an overview of the geological context, and a general description of mineralization framed into the host tectonic units recognized in the literature.

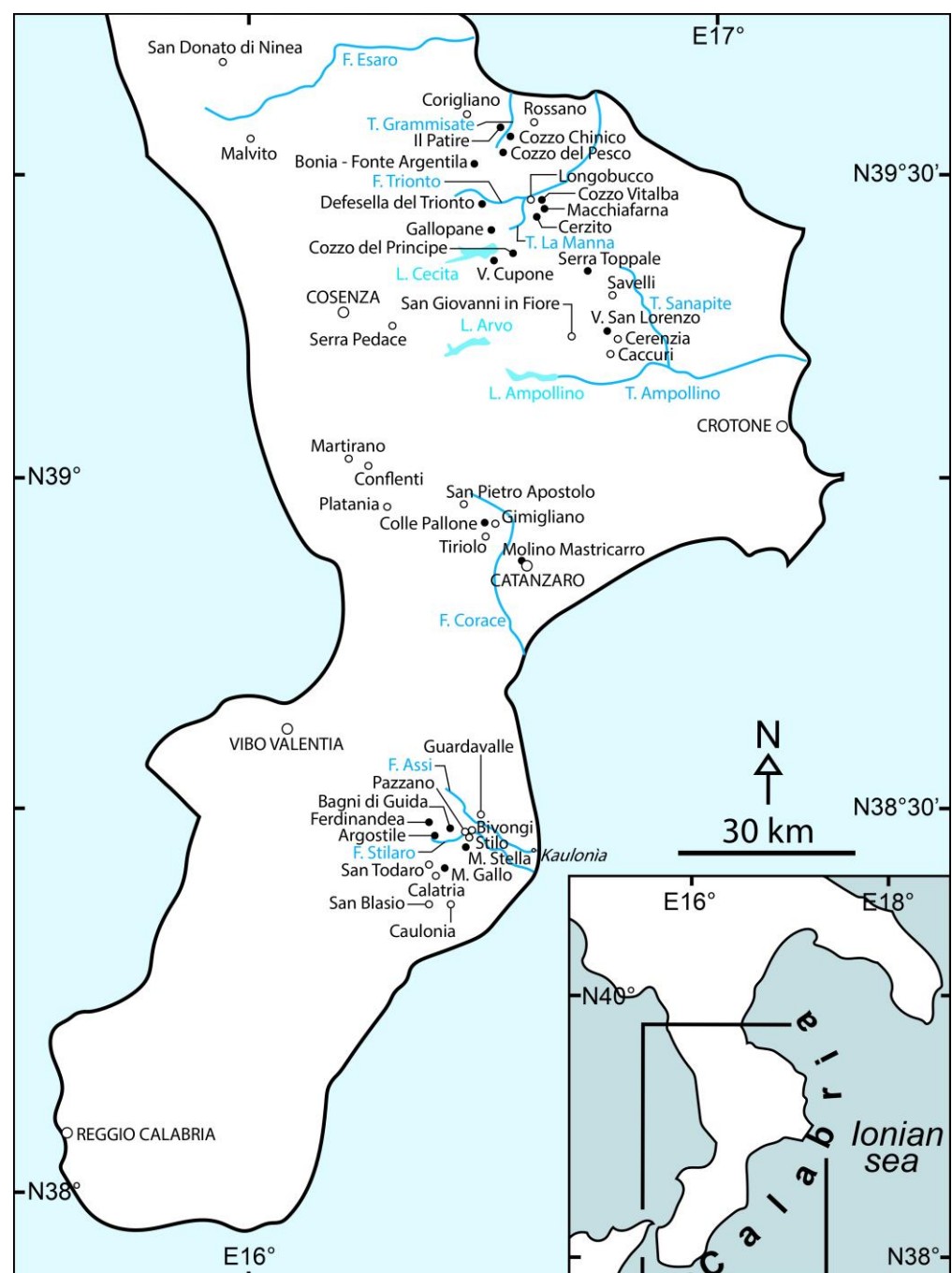

**Figure 1.** Geographic sketch-map of Calabria with toponyms of the areas characterized by mineralization.

## 2. Geological Setting

The Calabria-Peloritani terrane [58], i.e., the "calabro-peloritano arc" [59], is a nappe pile stacked-up during the Alpine orogeny. Amodio-Morelli et al. [59] suggested the distinction of several Alpine tectonic units, which in the Sila and Serre Massifs of Calabria (Figures 2 and 3) are: the Gimigliano Unit, the Diamante-Terranova Unit, the Bagni-Fondachelli Unit, i.e., the Fiume Pomo Unit [60], the Polia-Copanello Unit, the Monte Gariglione Unit, the Longobucco-Longi-Taormina Unit, and the Stilo Unit.

The complex of ophiolite tectonic units (Gimigliano and Diamante-Terranova units) affected by high pressure–low temperature (HP-LT) metamorphism (e.g., [61]) underlies the Fiume Pomo Unit, which, in turn, lies below the Castagna Unit (e.g., [62–64]).

The Fiume Pomo Unit is characterized by phyllites equilibrated during the Variscan orogeny under greenschist facies conditions [65]. The Castagna Unit is dominated by orthogneisses (mostly augen gneisses), minor muscovite-leucocratic gneisses, paragneisses, actinolite schists, and amphibolites, that equilibrated under amphibolite facies conditions during the Variscan orogeny [65,66]. Later, these two units were locally affected by the Alpine tectono-metamorphic overprint, especially along ductile shear zones [65–67].

Some authors [68–70] concluded that the metamorphic and intrusive successions within the Polia-Copanello, Monte Gariglione, Longobucco-Longi-Taormina, and Stilo units belong to a nearly continuous late Variscan continental crust block. Accordingly, Festa et al. [71] proposed a simplified tectonic scheme with this continental crust block belonging to a single large Alpine nappe, i.e., the Sila-Serre Unit (Figures 2–4); this nappe overthrusted the complex of ophiolite units and the Fiume Pomo and Castagna ones (e.g., [66,72], and references therein).

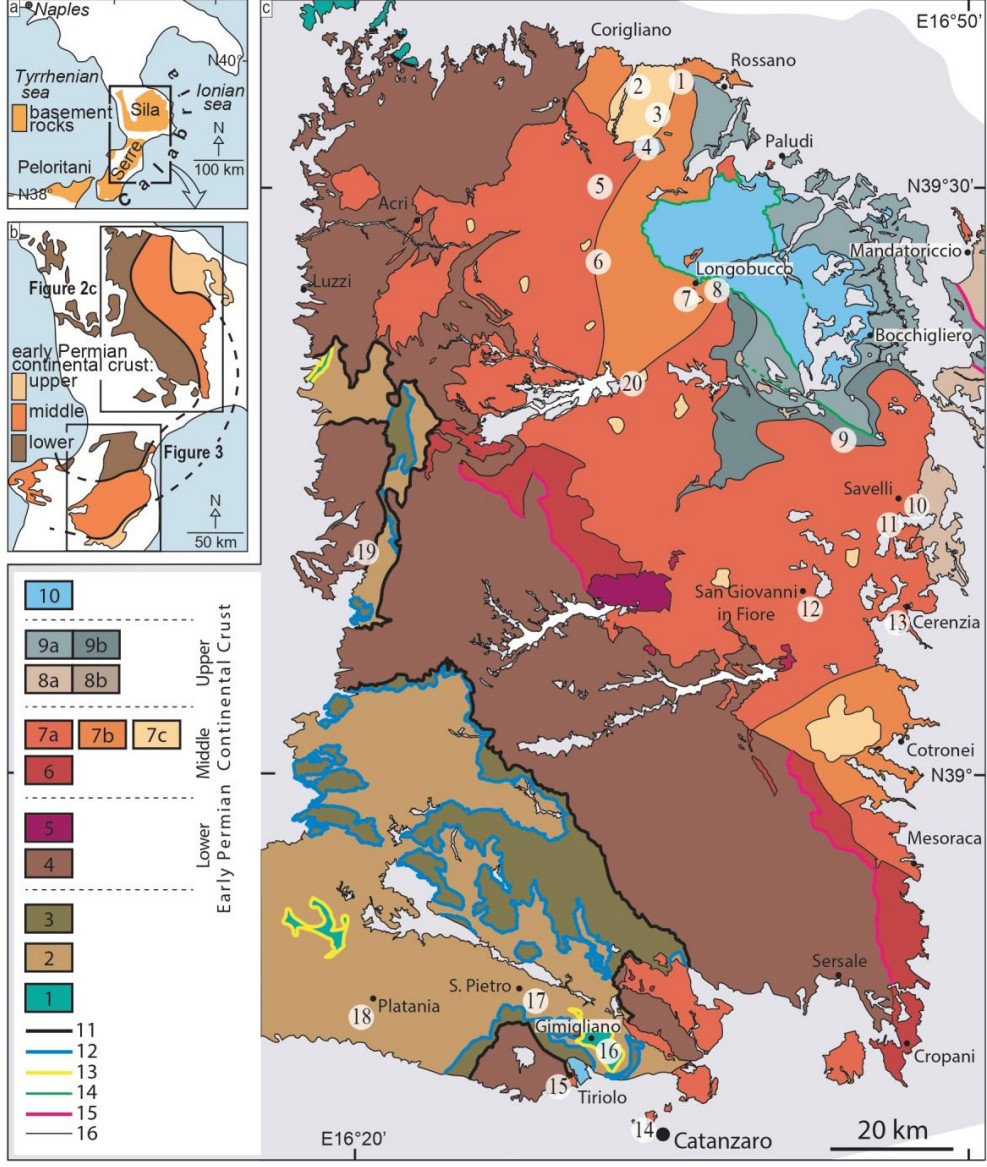

**Figure 2.** (**a**) Sketch map showing the distribution of basement rocks exposed in Southern Italy (modified after [71]); (**b**) sketch map of the lower, middle, and upper late-Variscan continental crust

exposed in Calabria (modified after [68]); (**c**) geological sketch-map of the Sila Massif area with numbered location (numbers 1–20 in the white circles are the same as reported in Table A1 and in paragraph 4) of the pertaining mineralization sites (modified after [73–76]). 1—Ophiolite units (e.g., Gimigliano and Diamante-Terranova units). 2—Fiume Pomo Unit (former Bagni-Fondachelli Unit; dominant phyllites). 3—Castagna Unit (dominant augen gneisses). Sila-Serre Unit: 4—former Monte Gariglione and Polia-Copanello units (granulites, gneisses, paragneisses, migmatites); 5, 6, 7a–c—Sila and Serre batholiths (i.e., 5—norite-diorite; 6—tonalites; 7a—granodiorite; 7b—Bt-Ms cordierite-bearing granodiorite-monzogranite; 7c—Bt-Ms-And-Sil-Crd-bearing granodiorite-monzogranite, Bt-Ms-And-Sil-Crd leucogranite, and small fine-grained granite); 8a—Mandatoriccio/Mammola Unit (dominant schists); 8b—contact aureole (spotted schists and fels at the expense of the Mandatoriccio/Mammola Unit schists); 9a—Bocchigliero/Stilo-Pazzano Unit (dominant phyllites); 9b—contact aureole (spotted schists and fels at the expense of the Bocchigliero/Stilo-Pazzano Unit phyllites); 10—Longobucco/Stilo cover (dominant carbonates). Main Alpine thrust at the base of the: 11—Sila-Serre Unit; 12—Castagna Unit; 13—Fiume Pomo Unit. 14 –Minor Alpine thrust and faults. 15—Hercynian shear zone. 16—Lithostratigraphic contact.

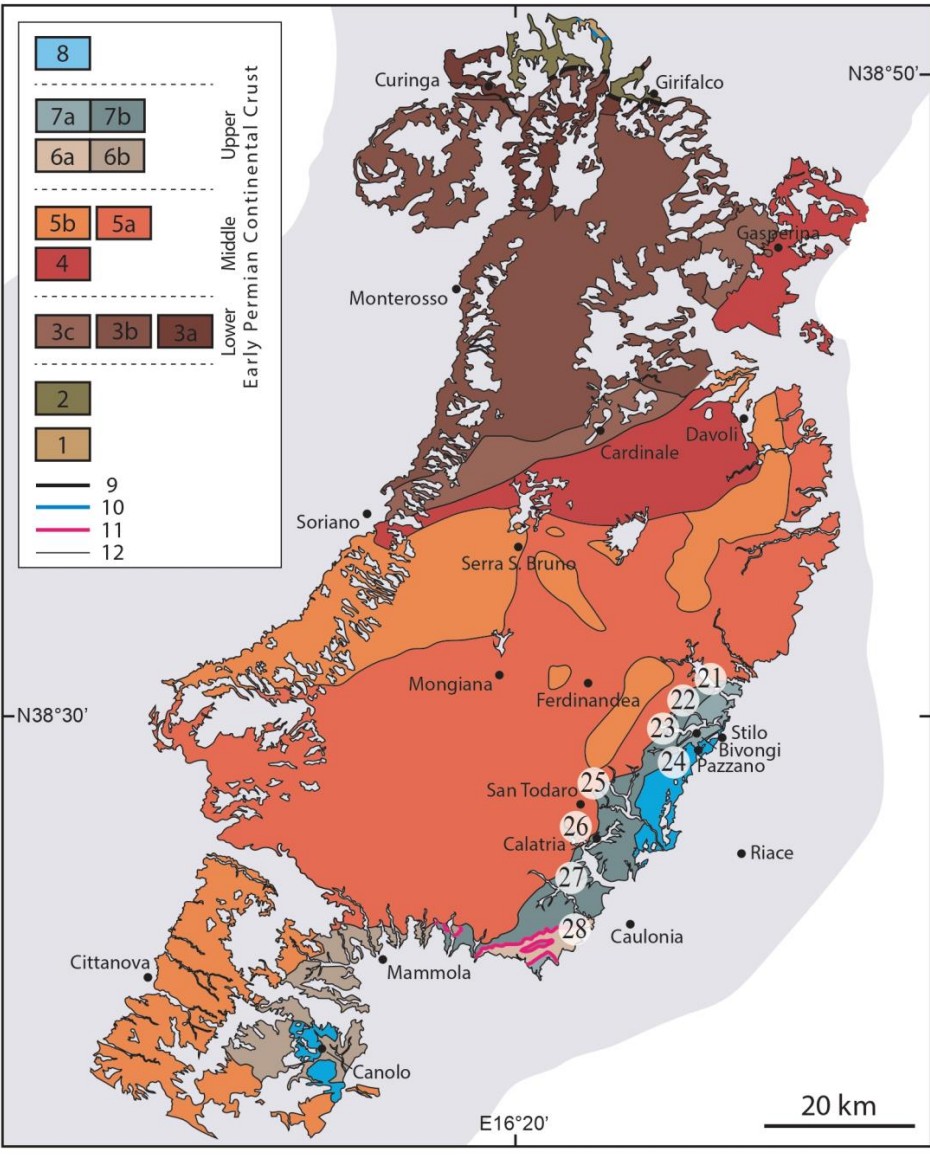

**Figure 3.** Geological sketch-map of the Serre Massif area with numbered location (numbers 21–28 in the white circles are the same as reported in Table A2 and in paragraph 5) of the pertaining mineralization

sites (modified after [74,76]). 1—Fiume Pomo Unit (former Bagni-Fondachelli Unit; dominant phyllites). 2—Castagna Unit (dominant augen gneisses). Sila-Serre Unit: 3—former Monte Gariglione/Polia-Copanello Unit (i.e., 3a—felsic and mafic granulites; metapelites and metagabbros; 3b—migmatitic metapelites; 3c—migmatitic border zone); 4, 5a,b—Sila and Serre batholiths (i.e., 4—tonalites; 5a—granodiorite; 5b—Bt-Ms cordierite-bearing granodiorite-monzogranite); 6a—Mandatoriccio/Mammola Unit (dominant schists); 6b—contact aureole (spotted schists and fels at the expense of the Mandatoriccio/Mammola Unit schists); 7a—Bocchigliero/Stilo-Pazzano Unit (dominant phyllites); 7b—contact aureole (spotted schists and fels at the expense of the Bocchigliero/Stilo-Pazzano Unit phyllites). 8—Longobucco/Stilo cover (dominant carbonates). Main Alpine thrust at the base of the: 9—Sila-Serre Unit; 10—Castagna Unit. 11—Hercynian shear zone. 12—Lithostratigraphic contact.

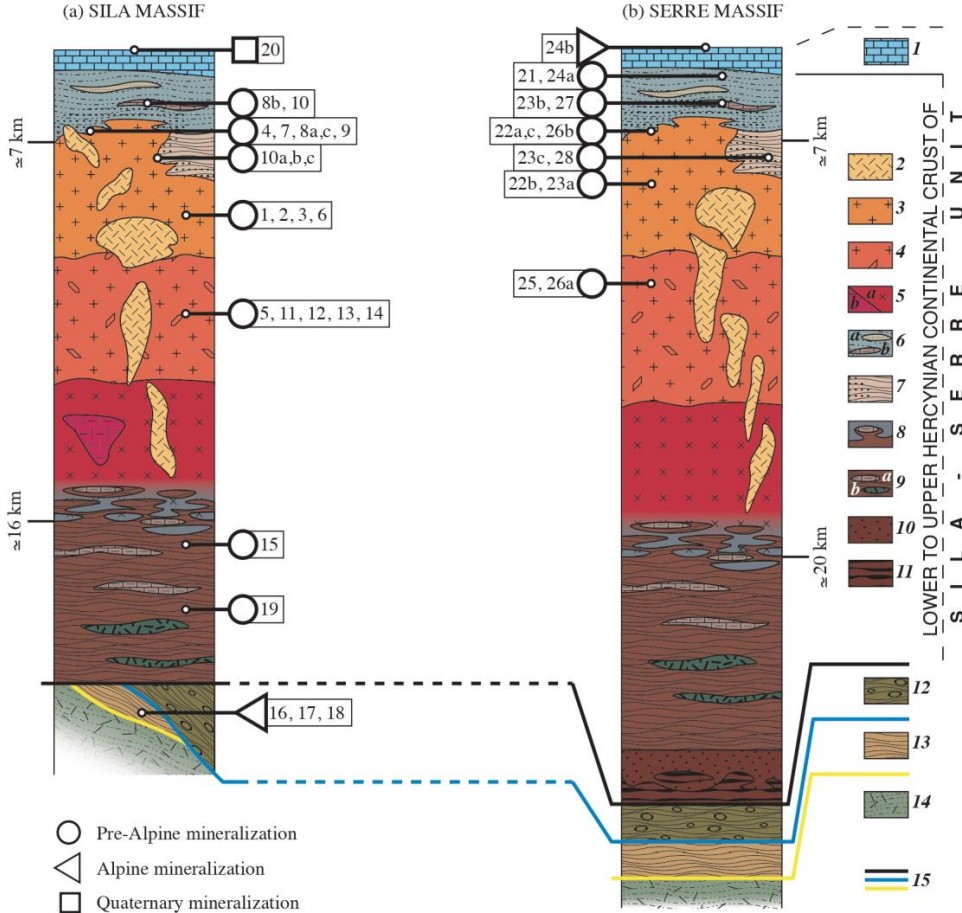

**Figure 4.** Schematic stratigraphic columns of the Sila-Serre Unit, in the Sila (**a**) and Serre (**b**) areas (modified after [71]), with numbered location of pertaining mineralization sites (numbers 1–28 in the squares are the same as reported in Tables A1 and A2, and in paragraphs 4 and 5) framed in the different structural levels; the Alpine Castagna, Fiume Pomo and Ophiolite units are also reported. 1—Longobucco/Stilo cover. 2—Peraluminous granites. 3—Granodiorites and granites. 4—K-feldspar megacryst-bearing granodiorites. 5—Quartz-diorites to tonalites. 6—Slates, phyllites with interbedded (**a**) marbles and (**b**) metavolcanites of the Bocchigliero and Stilo-Pazzano units, derived from Cambrian to Carboniferous protoliths (dots indicate the metamorphic contact aureole). 7—Micaschists of the Mandatoriccio and Mammola units (dots indicate the metamorphic contact aureole). 8—Migmatitic border zone. 9—Migmatitic paragneisses with interbedded (**a**) marbles and (**b**) metabasites. 10—Felsic granulites. 11—Layered metagabbroic rocks. 12—Castagna Unit. 13—Fiume Pomo Unit. 14—Ophiolite units. 15—Alpine tectonic contacts (line colors as in Figures 2 and 3).

The almost complete late Variscan continental crustal block shows a thickness up to about 23 km [69,70,77–79], consisting of three crustal levels: lower, middle, and upper (Figure 2b). The lower crust (formerly assigned to the Monte Gariglione and Polia-Copanello units) shows a thickness of about 7–8 km, and is composed of mafic and felsic granulites, below, and of amphibolite-granulite facies migmatitic paragneisses, above (e.g., [70,80–82]), juxtaposed by a shear zone [83].

The late Variscan crustal block continues upwards with the up to about 12 km thick Sila and Serre batholiths [84,85] that occupy the middle crust (Figures 2–4) [70,77,78]. The main magma affinity of the batholiths is calc-alkaline with isotopic signatures, indicating mantle-crust mixed origin [85–88].

The Sila and Serre batholiths mostly consist of late Carboniferous-early Permian tonalites, quartz-diorites, and two-mica peraluminous granites, biotite granodiorites and occasional small bodies of gabbro-diorite (Figures 2–4); sporadic volumes of aplite-pegmatite dikes are present especially along the roof of the batholiths [84,85,89–91], and within the related upper crust host rock (e.g., [92,93]). A wide migmatitic border zone is observed in-between batholiths and lower crustal migmatitic paragneisses; in the Sila Massif, the contact is characterized by an amphibolite-granulite facies shear zone, with supra- to sub-solidus deformation conditions [73,94,95]. The contact between granitoids and the upper crustal host rock is sharp and marked by an extensive metamorphic aureole (Figures 2–4) [92,93,96–98].

The upper crust consists of Variscan tectono-metamorphic units composed of amphibolite facies metapelites and meta-greywackes (i.e., Mandatoriccio and Mammola units), and a succession of phyllites, meta-carbonates, and acidic to mafic metavolcanites metamorphosed under sub-greenschist to greenschist facies conditions (i.e., Bocchigliero and Stilo-Pazzano units) [74,96–99]. It is worthy to underline that these Variscan tectonometamorphic units, together with Sila and Serre batholiths, were formerly ascribed to the Longobucco-Longi-Taormina and Stilo units [59].

Magmatic products, consisting of porphyritic, mafic and felsic dikes and sills, with rhyolitic to dacitic, and andesitic compositions, intruded the Sila and Serre batholiths, and the upper crust host rock [100,101]. This calc-alkaline to alkali-calcic affinity magmatism occurred mostly during the early Permian (between 295 and 277 Ma) crust exhumation and transtentional tectonics [100].

Finally, a Mesozoic cover, dominated by carbonate rocks, discordantly overlies the crystalline bedrock (e.g., [102–104]) and represents the so-called "Longobucco cover", in the Sila Massif [75], and the "Stilo cover", in the Serre Massif, as the upper part of the Sila-Serre Unit [71] (Figures 2–4).

## 3. Literature Background on Polymetallic Mineralization

In this section, we report the literature background on polymetallic mineralization in the Sila and Serre Massifs, summarized in Tables A1 and A2 following a temporal order, from pre-Alpine to post-Alpine, and listed with progressively numbered locations and short informative descriptions. The same numbered locations are indicated in the schematic geologic maps (Figures 2 and 3) and cross-sections (Figure 4).

A few pioneering papers describing the mineralization in Calabria were largely based on field observations, not supported by the analytical characterization of minerals. In these papers, two kinds of approaches were adopted: (i) descriptive, focussed on an area with several mineralization sites close to each other (e.g., [36,37]); (ii) lithostratigraphic, where several mineralization sites (scattered in areas far from each other) were framed in the same host tectonic unit (e.g., [53–55,57]). This second approach gives a global view highlighting possible genetic links between single mineralization sites, and allows immediate identification of inconsistencies with the updated literature on orogenic events recorded in Calabria's crystalline basement. Specifically, according to De Vivo et al. [53], every mineralization is hosted by units of continental derivation, and most of them would be genetically linked to the late-Variscan magmatic intrusions; mineralization occurrences related to sedimentary and metamorphic processes were found only in a few cases. These

authors observed differences in the mineralization occurring in the Longobucco, Monte Gariglione, and Stilo units, all affected by late-Variscan magmatic intrusions. In the Longobucco Unit, polymetallic Pb-Zn-Cu-Fe-As mineralization (sphalerite, galena, pyrite, arsenopyrite, chalcopyrite, marcasite and malachite) were reported, with calcite, quartz and fluorite as gangue minerals. They were mainly found in the contact aureole developed in the phyllites of the pre-Alpine Bocchigliero Unit, especially at the top of the Sila batholith, where the effect of the contact metamorphism is more intensive. Moreover, primary uranium mineralization within granites of the Longobucco Unit was hypothesized, based on the finding of secondary autunite and torbernite deposited in the nearby lacustrine sediments. Differently, in the Monte Gariglione Unit, the usual metalliferous minerals were not found; this unit would be characterized by the presence of primary accessory minerals (monazite and zircon) containing more than 6.35% $ThO_2$ [39,40,44,53,105]. In the Stilo Unit, Mo-mineralization (veinlets of molybdenite with quartz as gangue mineral, found in the Serre area), and Ba-mineralization (baryte found in the Sila and Serre areas), occurred within the late-Variscan granites; additionally, Pb-Zn-Fe-Ag-As-Cu mineralization (sphalerite, galena, pyrite, arsenopyrite, and chalcopyrite) were found in the granitic and metamorphic rocks adjacent to the intrusions [53].

In line with De Vivo et al. [53], Bonardi et al. [54] argued that, apart from a few recent mineralizations, mainly pre-Alpine mineralized systems occurred in Calabria; most of them were split-up during the Alpine evolution. These authors took into consideration metallic and not-metallic mineralization as well, and proposed some genetic hypotheses, by distinguishing them into: (i) pre-Alpine, stratabound and vein-type, within metamorphic and igneous rocks; (ii) Alpine, mainly stratabound, within metamorphic and sedimentary rocks; (iii) recent, of alluvial type [54]. In more details, the pre-Alpine stratabound mineralization, represented by magnetite, graphite and sulphides (i.e., pyrite, chalcopyrite, galena, sphalerite and pyrrhotite), are found within the metamorphic rocks of the Bagni, Polia-Copanello and Stilo units, whereas the vein-type sulphide (arsenopyrite, pyrite, chalcopyrite, sphalerite and pyrrhotite) mineralization occurs in the Stilo and Polia-Copanello units. The pre-Alpine mineralization hosted by igneous rocks consists mainly of sulphides (molybdenite, sphalerite, galena, pyrite, chalcopyrite, pyrrhotite, arsenopyrite), and subordinately of baryte, fluorite and U-bearing minerals, occurring within the Longobucco, Stilo and Polia-Copanello units. It was hypothesized that the pre-Alpine mineralization within igneous rocks were connected to a Variscan porphyry copper-molybdenum system, subsequently split-up during the Alpine tectonic evolution [54]. However, all the genetic hypotheses put forward by these authors were not supported by any analytical investigation of the minerals, nor by the few fluid inclusion studies available for the Calabrian mineralization [106–108]. Concerning the Alpine mineralizations, they were reported within the San Donato Unit (baryte, cinnabar and galena), in the ophiolitic units derived from Jurassic-Cretaceous oceanic crust (the Gimigliano, Malvito, Diamante-Terranova units, with pyrite and chalcopyrite mainly found in the Catena Costiera territory), and within the clastic sediments of the Miocene deposits (Th, Mn, S, NaCl and lignite). Finally, the recent mineralization would be constituted by the radioactive ones of secondary origin occurring in the Longobucco area [54].

Later, Lorenzoni et al. [55] examined two possible main factors controlling the metallogenesis and distribution of mineralization in the different tectonic units: (i) different types of metamorphism; (ii) magma emplacement depth and type of magmatic differentiation. According to Colonna and Zanettin-Lorenzoni [109], the Variscan metamorphism that affected the upper crustal rocks of the Bagni Unit did not control neither the distribution nor the concentration of ore minerals such as pyrite, but this would rather reflect primary sedimentary concentrations. On the contrary, Lorenzoni et al. [55] suggested that intensive metasomatic effects and mobilization producing metal concentration (mostly iron) took place in rocks of the lower and intermediate crust, during amphibolite-granulite facies Variscan metamorphism. Concerning the subsequent Alpine metamorphism, the Bagni Unit would have been only marginally affected, with no evident modification of

the mineralization that was already present at the end of the Variscan orogenesis [55]. In contrast, the Castagna Unit, consisting of the pre-Alpine gneisses intruded by granodiorites, and pertaining aplite-pegmatites, would have suffered a very intense Alpine dynamic metamorphism [55]. Here, Fe-Cu sulphides mineralization linked to aplite-pegmatites were occasionally found in the Castagna Unit, only where the primary lithotypes were preserved with respect to the Alpine tectono-metamorphic events [55]. This observation led the authors to infer that primary concentrations of ore minerals were dispersed by the subsequent Alpine metamorphism, and that the scarcity of ore mineralization in the Alpine ophiolitic units might be partly related to this effect, as in the case of the Gimigliano Unit, where only a few lenses of ore minerals were found within metabasites [55]. The spatial relationship existing between most of the Calabria ore mineralization and the late Variscan intrusions was observed by many authors (e.g., [53,55]), even though their genetic association is still to be proved. Investigating the role of intrusion depth and the degree of differentiation, it was observed that radioactive minerals were linked to the more differentiated intrusions, whereas Fe-Pb-Zn-As-Cu sulphides were related to weakly differentiated zones [55]. It was concluded that anatectic melts could not produce ore concentrations in the lower crust, whereas the late Variscan magmas with a mixed mantle-crust source promoted concentrations of polymetallic mineralization in the deeper granodiorite, and in the roof of the shallow two-micas granite and granodiorite [55].

Finally, Zanettin-Lorenzoni and Zuffardi [56] considered the Calabria-Peloritani Arc as a well-defined metallogenic province, which is characterized by some similarities with Sardinia. These authors also considered it as a barren region in the metallogenic maps of the Mediterranean area, because of the small size of its mineralization despite their numerous occurrences [56].

## 4. Mineralizations in the Sila Massif in an Updated Geological Framework

In this section, the scientific literature on three major areas of historical mining interest in the Sila Massif is reviewed: (i) Longobucco-Corigliano-Rossano; (ii) San Giovanni in Fiore-Savelli-Cerenzia; (iii) Catanzaro-Tiriolo-Gimigliano. Data on geochemical anomalies for specific metal elements in some localities [46,47,51,107] are detailed. Limited information from the literature on the amount of metallic minerals once extracted are also reported for some mining sites. The list of mineralization sites in the Sila Massif is shown in Table A1, where we report a summary of the available literature information and, in the last column, a review of the occurrences of mineral deposits with respect to the current tectono-magmatic and -metamorphic units. The same mineralization sites are reported on the geological map (Figure 2c) and the stratigraphic column (Figure 4a).

### 4.1. Longobucco-Corigliano-Rossano Area

4.1.1. Longobucco (CS) (Sites n. 6,7,8a,8b,8c,20; Table A1; Figures 2c and 4a)

Several remarks about ancient mines once active in the Longobucco area are reported in the oldest literature (e.g., [7,8]), even though in more recent times traces of those mines were hardly found [37]. Argentiferous galena, bournonite and arsenopyrite characterized this area, with a higher Ag-content of galena hosted by phyllites, which form the granite country rocks in the area [7]. Longobucco was historically known especially for the extraction of silver from argentiferous galena, with an annual average production of 87 kg of Ag, from 1268 to 1277. In the first half of the 16th century, the Argentera of Longobucco was considered the most important silver mine; on 1645, the excavation of a galena dike close to the Trionto river produced just 4 parts of silver over 100 parts of galena (e.g., [3,110]). The Longobucco mineralizations were prospected and exploited between 1723 and 1783, and at the beginning of the 19th century, when Ag-Pb-Sb $\pm$ Cu-Fe were extracted [8,9,111]. From 1828 to 1832, the "Acqua di Radica" site produced 1800 "cantari" of Pb [17]. Anyway, galena was hardly found in the Longobucco area. At the end of the 19th century, the finding of some galena within landslide debris at Spartari and along the Macrocioli river was reported, but a galena ore body was not observed [23]. Indeed, foundries for the extraction

of Ag from argentiferous galena were located just along the Macrocioli river [110]. For the Longobucco area, positive geochemical anomalies of Zn-Fe-Sb (neither Pb nor Ag) were detected in soils [107].

Ore bodies hosted by magmatic rocks (granite, quartz-monzonite, porphyritic and quartz-monzonitic dikes), and metamorphic rocks (schists and hosted marble lenses) were studied [31,32,34,35,37]. A lower sulphides content was reported for dikes hosted by phyllites, with respect to those within granitic rocks [37,53]. Multiple fracturing and mineralization episodes were recognized, producing the following paragenetic sequence [37]: calcite I + sphalerite; quartz + pyrite (± marcasite); calcite II; galena + chalcopyrite; (secondary) calcite III. Sphalerite is the most abundant ore mineral [53], whereas galena was sporadically found in small veins filling fractures within calcite II [37]. Noteworthy, galena mineralization was reported from quartz veins, hosted by granitic rocks close to the contact with Lias carbonate rocks. This mineralization was considered as related to post-Triassic porphyritic dikes, and of Mesozoic (post-Lias) age [29]. However, dike magmatism in the area is well constrained to the late Carboniferous–early Permian age [100]. Therefore, these carbonate rocks would be interbedded in the Variscan Bocchigliero Unit succession [99].

Numerous mineralized sites are reported in the literature for this area (Table A1; [54]). In particular, Pb-Zn ± Fe-Cu mineralization (sites n. 6, 7, 8a) of suggested hydrothermal origin were observed in veins within Variscan granitoids, close to the contact with phyllites of the Longobucco Unit [54]. In addition, pyrite vein mineralization within carbonatic lenses intercalated to phyllites of the Longobucco Unit was found at Cozzo Vitalba (site n. 8b) [53]. However, it should be specified that these phyllites would belong to the Variscan Bocchigliero Unit, i.e., the late Variscan upper continental crust within the Alpine Sila–Serre Unit [71]. Moreover, the presence of scheelite (W) mineralization in skarns within schists of the contact aureole of the Sila batholith at Croce Reinella (site n. 8c) was documented [107,112,113]. Their genesis was related to the effect of late-Variscan intrusions on pre-existing host rocks [112]. Furthermore, secondary U-minerals autunite and torbernite were found at Vallone Cupone, Cozzo del Principe and Gallopane (site n. 20), within lacustrine sediments overlying magmatic rocks of the Longobucco Unit [54,55].

4.1.2. Corigliano-Rossano (CS) (Sites n. 1–5; Table A1; Figures 2c and 4a)

The presence of Zn-Pb-Fe mineralization considered of hydrothermal origin, occurring north of Longobucco (sites n. 1–5; Table A1; Figure 2c), mostly in veins within the Variscan granitoids close to the contact with phyllites of the Longobucco Unit, is also reported [54,55]. According to the recent literature [100], the phyllites formerly attributed to the Longobucco Unit should belong to the Bocchigliero Unit.

Among the excavation sites for argentiferous galena, the site "Argenteria", located north of Longobucco towards Corigliano, was reported [17]. In the locality "Baracconi di Campagna—Fonte Argentila" (site n. 5), galena is abundant, possibly associated to minor pyrite in a quartz gangue [110]. Mining tunnels once active at this site for the extraction of Pb are documented [110]. Indeed, positive geochemical anomalies of Pb were detected in this area [47,51]. At Torrente Petraro, strictly close to Fonte Argentila (site n. 5), the mineralizing fluids would have circulated through highly deformed rocks, leading to the formation of mineralized lenses, which are mainly located at the intersection of important fault systems [39,53].

Mineralized veins (sphalerite, galena, pyrite, malachite, and quartz) hosted by granitic rocks of the Longobucco Unit would be located close to the Norman monastery Santuario del Patire (site n. 2) in the Corigliano territory [54]; this site was indicated as one other probable source area for the argentiferous galena [110]. Positive geochemical anomalies of Rb, Cs, Sr, Ta, U, Th, Ce, La, Ba, Nb and Au were detected in the surrounding area of the Patire site [107].

At the mineralized site Cozzo del Pesco (n. 4) in the Corigliano area, pyrite is the most abundant ore mineral, in a quartz gangue, with minor sphalerite veinlets within

granodiorites. Porphyry, aplite, pegmatite and microgranite dikes are common at this site [39,53].

In the Rossano area, the mineralization sites reported in the literature are: (i) Torrente Grammisate (site n. 1), where a 30 cm wide mineralized dike (sphalerite, galena, chalcopyrite, fluorite) hosted by granitic rocks of the Longobucco Unit was found at 250 m a.s.l [39,53,55]; (ii) Cerasaro (site n. 3), located to the south-east of Cozzo Chinico, where a 4 m wide dike mineralized by pyrite in quartz gangue was found hosted by granodiorites [39,53–55]. The Rossano area would be also characterized by the presence of W-Zn-Cu skarns within the contact aureole of the late-Variscan granitoid, similar to those observed at Longobucco and Savelli [112]. Positive geochemical anomalies of Pb, Zn, Cu, W and Sn were detected in the Rossano area [47].

The mineral association pyrite, chalcopyrite, sphalerite, galena, arsenopyrite, pyrrhotite, electrum, Bi-tetrahedrite, native bismuth, Bi-Ag sulfosalt (pavonite type) and possibly montroydite, in a dominant quartz gangue, characterizes mineralized Fe-Cu-Zn-Pb-As-Bi-Au-Ag-bearing vein samples from an undefined area to the west of Rossano [108]. Two mineralization stages were identified: (i) pyrite + arsenopyrite + base-metal sulphides + quartz; (ii) Bi-tetrahedrite + pavonite + native bismuth + electrum + quartz. The element Au was hosted by arsenopyrite, and mostly by electrum, the latter typically occurring along fractures and in nuggets within pyrite. The first mineralization stage took place under a temperature range of 300–350 °C, and a pressure of 1.5 kbar, based on chemical analyses, whereas an upper thermal limit of 270 °C was inferred for the second stage from the presence of native bismuth [108]. Moreover, fluid inclusion analyses performed on stage II-quartz showed a 135–185 °C range for the minimum temperature of formation and a maximum temperature of 250 °C, with salinities of 6.8–7.3 wt% NaCl eq., for the second stage ore-forming fluid [108]. These results allowed the authors to trace some similarities with gold-bearing veins closely related to late-Variscan granitoids of the French Massif Central [108].

*4.2. San Giovanni in Fiore-Savelli-Cerenzia Area*

4.2.1. San Giovanni in Fiore (CS) (site n. 12; Table A1; Figures 2c and 4a)

At the San Giovanni in Fiore site, galena and pyrite mineralization within granitic rocks, locally encrusted by meta-autunite, is documented [3,114]. Positive geochemical anomalies of Mn, Y, Cr, Fe, As, Co, Cu, Ni and Sb were detected to the north-east and east of San Giovanni in Fiore, within tonalite, granodiorite and monzogranite intrusions [107].

4.2.2. Serra Toppale-Savelli (KR) (Sites n. 9, 10a, 10b, 10c; Table A1; Figures 2c and 4a)

In the nearby of Savelli, an important mining site was located at Serra Toppale (n. 9), where galena, pyrite and sphalerite mineralized veins cross-cut Variscan granitoids, close to the contact with phyllites [54].

Moreover, in the Savelli area, the following mineralization are documented: (i) pyrite mineralization (site n. 10a), hosted by hornfels near the contact with granodiorites [39,53,55]; (ii) W-Zn-Cu skarns (site n. 10b), within contact aureoles of the late-Variscan granitoid intrusion, similar to those observed at Longobucco and Rossano [112]; (iii) at Torrente Sanapite (site n. 10c), chalcopyrite, malachite, galena and pyrite in pegmatitic dikes, injecting paragneisses near the contact with granodiorites [53].

4.2.3. Cerenzia-Castelsilano (KR) (Sites n. 11, 13; Table A1; Figures 2c and 4a)

The Cerenzia territory has been interested by poor minerary exploitation [53]. Mineralized dikes hosted by granitic rocks of the Longobucco Unit, mainly consisting of galena, sphalerite, pyrite, chalcopyrite and arsenopyrite ore minerals (site n. 13), with quartz, calcite and fluorite gangue minerals, were reported [39,52–54]. Anyway, prospection excavations conducted up to the second post-war period, led to the conclusion that this area was not of economic interest [52]. Moreover, a 200 m long molybdenite dike, hosted by granitic rocks of the Longobucco Unit, was reported at Vallone San Lorenzo (site n. 11), in

the Cerenzia-Castelsilano area [115], even though subsequent prospection studies were not able to confirm this finding [53].

### 4.3. Catanzaro-Tiriolo-Gimigliano Area

#### 4.3.1. Molino Mastricarro, Fiumarella (CZ) (Site n. 14; Table A1; Figures 2c and 4a)

A baryte mineralization outcropping along the left bank of the Fiumarella river at Molino Mastricarro (Catanzaro) was reported for the first time in 1878 [15,16], occurring within green and red porphyritic rocks [11,12]. At this site, numerous pegmatites, aplites and lamprophyres frequently crosscut the granitic and metamorphic (phyllites) rocks [36]. A large baryte vein up to 80 m in length and 3 m thick, possibly formed by mineralizing fluids circulating through principal joints within a granitic body, was described [36]. The same fluids also percolated the wall-rocks along fractures, producing several minor veinlets of baryte. All these rocks seem to have suffered intense modifications, not of supergene origin [36]. The minerographic study showed that: (i) the vein mainly consists of spathic baryte, which, in turn, contains nodules and small veinlets of galena, occasionally associated to chalcopyrite and quartz; (ii) the baryte mineralization should have preceded the galena-chalcopyrite-quartz one [36]. Other minor minerals (fluorite, anglesite, pyrite, cerussite, calcite, wulfenite, delafossite, covellite, malachite, azurite, limonite, prehnite, linarite and connellite) were also reported from this deposit [17,36,116–118]. Geochemical anomalies of W (>23 ppm) and Zn (>120 ppm) were found for this area [51]. The baryte mine was exploited by the end of the 1960′s and continued until the beginning of the 1980′s of the 20$^{th}$ century, with a production of 25,000 ton/year with a yield of 40% barite [106]; after that, the ore deposit was declared depleted. This mineralization was classified as vein-type pre-Alpine within granitic and porphyritic rocks of the Stilo Unit [54]. The phyllites formerly assigned to the Stilo Unit, should be included in the Bocchigliero Unit [100].

A pioneering fluid inclusion study was performed on the baryte samples from the Molino Mastricarro mine [106], to ascertain whether this deposit could be classified as porphyry Cu-Mo type. The minerographic analysis confirmed the previously observed paragenetic sequence, i.e., the sulphides formed after baryte. The fluid inclusion analyses revealed a minimum range of temperature 190–235 °C, with a salinity of the mineralizing fluids 0–19.5 wt% NaCl eq., and an estimated average minimum pressure of 18.04 bars, for the formation of the baryte mineralization. Based on these results, the genesis of the Fiumarella deposit was put in relation either to epithermal precious metal (with base metals) vein deposits, or to the Kuroko type (i.e., volcanogenic massive sulphides) deposits; either way, excluding the porphyry Cu-Mo hypothesis [106]. This finding would differentiate the Molino Mastricarro mineralization with respect to other deposits occurring within plutonic rocks formerly belonging to the Stilo Unit (e.g., Bivongi in the Serre Massif, namely the Serre Batholith), which are characterized by the presence of molybdenite among their mineral association [54].

#### 4.3.2. Tiriolo (CZ) (Site n.15; Table A1; Figures 2c and 4a)

At the end of the 19th century, one of the most important mineral deposits of the Calabria region was located near the Tiriolo village. Fe-Cu-Zn-ore minerals (pyrite, chalcopyrite, sphalerite, limonite, tetrahedrite and bornite) were collected for the first time in 1878–1887 [15,17,19,20], associated to other minerals (calcite, spinel, garnet, vesuvianite, epidote, chlorite, malachite, azurite, aurichalcite, prehnite, brandisite, fassaite and mesolite) [15,17,19,20,24,25,119,120]. It is worth noting the presence of a green-blue coloured Zn- and Fe$^{3+}$-bearing spinel, as revealed by chemical, spectroscopic and X-ray diffraction studies [120–123], in accordance to the observed correlation existing between the blue-green colour and the Fe$^{3+}$-content of Zn-bearing spinels [124,125]. According to Lovisato [15,17,19], the major ore deposit is hosted by the Paleozoic limestones interleaved to schists, and in close relation to porphyritic magmatic intrusions. However, Panichi [24,25] just reported that the mineral deposit of Tiriolo was found "at the contact between crystalline and overlying carbonate rocks". According to De Angelis et al. [122], the ore deposit

is hosted by thermo-metamorphosed limestones close to granites and tonalites, cross-cut by porphyritic dikes. Following Crisci and Dattola [126], the ore would be hosted by calciphyres produced by thermometamorphism in the presence of fluids at the contact between the magmatic intrusion and pre-existing carbonate rocks.

### 4.3.3. Gimigliano (CZ) (Site n. 16; Table A1; Figures 2c and 4a)

In the Gimigliano area, a pyrite ore located along the slope of Colle Stretto and at the basis of Colle Pallone, in-between the Amato and Corace rivers, was reported [23,28]. Pyrite was embedded within quartzites intercalated with phyllite schists, and in masses up to 100 m wide and 7 m thick overlying the phyllites, in close relation to the granitic body [28]. A total amount of 100.000 tons of pyrite, with a sulphur content of 47%, was estimated for this ore deposit [28]. Excavation and extraction activities in this area were started in the 1920′s, with the main works being located at the site Cozzica (along the slope of Colle Stretto), and continued up to 1946 [2,28,53]. Concerning the genesis of this pyrite ore, different hypotheses have been put forward: (i) it might be strictly linked to the granite intrusions, as it was already observed in the case of some other sites in Calabria, such as Pazzano, where pyrite is found between limestones and schists close to the granite intrusion [28,29]; (ii) it might be linked to sedimentary and metamorphic processes [53–55,109].

In the Gimigliano area, a mineralized site at the Acqua Bollita locality was also reported, which is characterized by the presence of a ferruginous-arsenical water coming from phyllite schists hosting As-bearing pyrite, galena and stibnite mineralization [14,15,18,23].

## 5. Mineralizations in the Serre Massif in an Updated Geological Framework

In this section, the scientific literature on two major areas of historical mining interest in the Serre Massif is reviewed: (i) Stilo-Pazzano-Bivongi; (ii) San Todaro-Calatria-Caulonia. Data on geochemical anomalies for specific metal elements in some localities [47,51] are detailed. Some information from the literature on the amount of metallic minerals once extracted was also reported for some mining sites. The list of mineralization sites in the Serre Massif is shown in Table A2, where we report a summary of the available literature information and, in the last column, a review of the occurrences of mineral deposits with respect to the current tectono-magmatic and -metamorphic units. The same mineralization sites are indicated on the geological map (Figure 3) and stratigraphic column (Figure 4b).

### 5.1. Stilo-Pazzano-Bivongi Area (Sites n. 21, 22a, 22b, 22c, 23b, 24a, 24b; Table A2; Figures 3 and 4b)

Positive geochemical anomalies of Pb, As, Zn and Mo were found to the west of the Stilo village [47], where Fe-Ag-Pb-Au-Mn-Cu ore deposits were reported (e.g., [26], and references therein). Particularly noteworthy for economic issues, the limonite mine near to the Stilo and Pazzano villages (site n. 24b) produced 40–45% of Fe on average, which was worked at the metallurgical plant of Ferdinandea [17]. The limonite-goethite-hematite mineralization mainly occurs along the contact between Paleozoic phyllites and Mesozoic limestones (between Monte Stella and Monte Gallo localities), and also fills fractures that cut Jurassic to Cretaceous limestones. This mineralization would be of sedimentary/supergene origin, derived from the oxidation of pre-existing minerals (pyrite, sphalerite, chalcopyrite and galena), and Alpine in age [28,38,54,55].

Differently, according to Vighi [38], the sulphide mineralization hosted by granitic rocks and Paleozoic phyllites, cropping-out along the whole eastern border of the Serre Massif, would be late Carboniferous in age, because it is possibly genetically linked to the Variscan magmas emplacement.

In the area of the Pazzano and Bivongi villages, molybdenite mineralization is primarily hosted by granitoid rocks ([38], and references therein), and in some cases also by the nearby phyllite schists [23,26]. One of the major mining sites in this area was Cantiere Giogli (site n. 22b), which was exploited from 1937 to 1943, and was characterized by a molybdenite vein-type mineralization in a quartz gangue, with minor pyrite, chalcopyrite,

limonite, covellite, U-molybdate and ferrimolybdite [26,38,127–129]. The mineralized dikes (up to 25 cm thick) were considered of pneumatolytic origin, formed at elevated temperatures, for several reasons: (i) they are interpenetrated with the host quartz-monzonite rock, with gradual transition; (ii) the molybdenite crystals within dikes are often more than 1 cm in size; (iii) small molybdenite crystals are found also within the quartz-monzonite host rocks, often far away from the mineralized dikes [38]. This ore was not considered of economic importance because of its low molybdenite content, i.e., about 5% [38]. According to Bonardi et al. [54], the pneumatolytic-hypothermal molybdenite-pyrrhotite mineralization hosted by granitic rocks of the formerly Stilo Unit indicates a deep origin within the continental crust, because the presence of Mo suggests assimilation of large portions of continental crust by the magma and associated fluids.

At the Argostile site (n. 22c), in the nearby of Monte Campanaro locality in the Stilo village area, a pneumatolytic mineralized vein (arsenopyrite, orpiment, quartz), hosted by quartz-monzonite close to the contact with the Palaeozoic schists, crops-out with a lenticular section up to 1 m wide [38]. For the same Argostile locality, a late Variscan galena-sphalerite-arsenopyrite mineralization of hydrothermal origin, hosted by granodiorites, is also reported [55].

At the Serravetta locality, close to the Bivongi village, a pyrite-chalcopyrite ± sphalerite mineralization occurs along the right bank of Torrente Pardalà (site n. 22a), as a diffused impregnation zone within the Palaeozoic schists, close to the contact with quartz-monzonite [38]. Old prospection galleries, nowadays collapsed and closed, were excavated at this site. Waste material from the mine, with 25% average sulphide content, was analysed: dominant pyrite of first generation, containing second generation minerals (chalcopyrite, sphalerite, calcite) infilling fractures, was observed [38]. Similarly, moving from Stilo towards the Guardavalle locality, a chalcopyrite-sphalerite-galena mineralization is located in the stream valley of Fiumara Assi (site n. 21), in the form of a diffuse impregnation zone hosted by the Palaeozoic phyllite schists [38].

In the Bivongi village area (site n. 23b), pre-Alpine mineralization within metamorphic rocks of the formerly Stilo Unit occur as small sporadic lenses of pyrite and pyrrhotite (with quartz gangue) hosted by carbonaceous phyllites, and as disseminations of chalcopyrite, pyrite and pyrrhotite within basic rocks intercalated with phyllites [54].

In the nearby of Monte Stella locality, in the Stilo village territory (site n. 24a), a dike filling a fracture within Palaeozoic schists, and located at about 2 km from the contact with the granite intrusion, is mineralized by sphalerite, galena and chalcopyrite in quartz gangue, with up to 70% estimated ore mineral content [38].

Finally, geochemical anomalies of Zn (>120 ppm) and As (>93 ppm) were reported [51] in the Monasterace village territory, in-between the Fiumara Assi and Fiumara Stilaro stream valleys.

### 5.2. San Todaro-Calatria-Caulonia Area (Sites n. 25, 26a, 26b, 27, 28; Table A2; Figures 3 and 4b)

Of particular note, Mo anomalies (>15 ppm) were found in the Serre Massif to the west of the Monasterace village within granodiorites, indicating the presence of Mo-mineralization [51]. Indeed, a molybdenite vein-type mineralization hosted by quartz-monzonite was reported in an area located about 250 m to the north of San Todaro village (site n. 25) [38,54]. Several mining galleries were excavated at this site, which were already collapsed at the end of the 1940′s: the molybdenite content, found in samples from the waste material of the mine, was less than 5% [38]. Similarly, a few centimetres thick quartz-molybdenite dikes, hosted by heavily fractured and altered quartz-monzonite, were observed at about 250 m to the N-NW of the Calatria village (site n. 26a), inside mining galleries [38].

Moreover, Vighi [38] studied a mineralized site located at 1.1 km to the N-E of the Calatria village (site n. 26b), with a vertical dike within phyllites, close to the contact with granites. The paragenetic sequence: pyrrhotite, marcasite, pyrite + quartz, sphalerite + calcite + chalcopyrite + galena, calcite II + pyrite II + quartz II, was suggested for this

mineralization. A maximum temperature of about 450 °C was estimated for the marcasite formation [38]. Pre-Alpine stratabound chalcopyrite mineralization, as disseminated lenses hosted by carbonaceous phyllites of the formerly Stilo Unit, is reported at the Monte Granieri locality, in the Calatria village territory (site n. 27) [54].

Finally, an As-Cu-Fe vein-type mineralization (arsenopyrite, pyrite, and chalcopyrite, in a quartz gangue) was found at the San Blasio locality, in the Caulonia territory (site n. 28), hosted by metamorphic rocks close to the contact with the Variscan granodiorites of the formerly Stilo Unit [54].

In the need to review the tectonic units hosting the above-described mineralization in the Serre Massif, in an updated geological framework, we can observe that such mineralization is distributed within the intermediate and upper portions of the late Variscan continental crust block (Figure 4b), i.e., within the Sila-Serre Unit [71]. The mineralization at sites n. 22b, 23a, 25 and 26a (Table A2; Figures 3 and 4b), occurring in granitoids previously assigned to the Stilo Unit (e.g., [54,126]), can be located in the intermediate crust; they might be considered genetically linked to the late Carboniferous intrusive magmatic events, which gave rise to the Serre batholith (e.g., [78,91]). The mineralization at sites n. 21, 22a,c, 23b,c, 24a,b and 26b-28 (Table A2; Figures 3 and 4b), occurring in igneous and metamorphic rocks, formerly assigned to the Stilo Unit (e.g., [54]), are located in the upper crust. They might be related to hydrothermal activity and/or contact metamorphism at the roof of the Serre batholith (e.g., [92]), or to the late Carboniferous-Permian dike magmatism [93,101].

## 6. Discussion, Open Issues and Future Directions

Our overview of the known literature on the mineralization of the Sila and Serre Massifs allowed us to evidence some critical aspects, and also to highlight some open issues which are worth dealing with, in order to deepen the understanding on this topic.

First of all, the literature on this subject is definitely not exhaustive and mostly out-of-date, mainly consisting of a few old scientific papers (e.g., [56], and references therein), and some archeo-minerary papers from which it is not clear whether the reports of mineral findings are based on direct analytical evidence made by the authors or not (e.g., [110]). Difficulties may sometimes arise when trying to exactly locate the mineralized sites on the geological and topographical maps, just based on reports found in the old literature. This happens for many reasons, among which is the proliferation of several slight differences in the way by which the same toponym name has been transmitted through time in the different papers; moreover, the presence of geographically distinct sites having similar toponyms (e.g., Reinella, Reginella, Righinella, and Reghinella, in the Longobucco area) gives rise to further confusion. In some cases, field prospections intended to verify the early reports on mineralization occurrences did not confirm those notices [130,131]. In this respect, it is of particular interest that many indications of the presence of gold in several localities in the Calabria mineralizing province were not subsequently confirmed. This led to the suggestion that, in past times other minerals, such as pyrite and altered biotite, might have been misleadingly identified as gold [130]. In the same stream, the recurrent difficulty of finding the famous argentiferous galena, encountered by several authors in different mining sites of Calabria (e.g., [5,23,37]), is also worth exploring further and clarifying.

The main open question emerging from our overview is the need for a deeper analysis of the genesis of these ore deposits. According to the old literature, every mineralization of the Sila and Serre Massifs occurred within continental derived units, and most of them were genetically linked to the late-Variscan magmatic intrusions; mineralization occurrences related to sedimentary and metamorphic processes were sporadically found [53]. Two distinct mineralization episodes were suggested for the Serre Massif: (i) slightly later than the Variscan intrusion; (ii) linked to the Alpine orogenesis [38]. According to Vighi [38], a marked lateral zoning would characterize mineralization occurrences in the Serre Massif, with high temperature molybdenite-quartz dikes hosted by quartz-monzonite of the inner parts, and low temperature sphalerite-galena-chalcopyrite-quartz dikes occurring within phyllites of the outer parts; such lateral zoning would testify that the formation of these

mineral deposits closely followed the granitic intrusion [38]. To this concern, our review of the host tectonic units has allowed us to locate mineralization of the Serre Massif within a stratigraphic column (Figure 4b), from which it is apparent that the molybdenite ore deposits (e.g., sites n. 22b, 23a, 25, 26a; Table A2; Figure 4b) are deeper and located in the intermediate portion of the Variscan continental crust, whereas the other ones are located in the upper crust (Figure 4b).

According to Bonardi et al. [54], apart from a few recent mineralizations, mainly pre-Alpine mineralized systems occurred in Calabria, which would have been split-up during the Alpine evolution, and found in a fragmentary way in the present orogenic structure. Even though some genetic hypotheses have been put forward by a few authors in the early literature (e.g., [53,54]), such hypotheses cannot be considered exhaustive at all. Some similarities about the geological context in which the Sila and Serre mineral deposits formed can be derived just from the overview of the previous literature, such as the frequently observed close relation of the mineralization with the Variscan intrusive granitic bodies and with their associated late stage and shallower porphyritic dikes (e.g., [15,16,37,53,54]). In addition, an interesting correlation between mineralizations and the highly faulted structural setting of the host rocks appears when trying to locate the mineralization reports on the geological maps. Moreover, several polymetallic mineralizations of the Sila and Serre Massifs show some common features with specific European ore deposits, ascribed to a post-Variscan and late Carboniferous-Permian geodynamic context. These similarities are attributed to the variety of intrusive magmas which are associated to the mineralization (i.e., granites and monzogranites, expression of late S-type and K-type Variscan magmatism; leucogranites, lamprophyres and andesitic dikes, ascribed to late-Carboniferous to Permian A-type magmatism) (e.g., [132,133]). The chemical differences recorded in the Calabrian mineralization might possibly be the consequence of several overlapping genetic processes, even though some groups of ore deposits also present in other European contexts may be identified. In detail, the main Zn-Pb-Fe-Cu vein-type mineralization, found in the Longobucco, Rossano, Savelli and Stilo-Bivongi areas, is also observed in the French Central Massif [134,135], in the Freiberg District and Harz Mountains in Germany [136–139], in the Kutna Hora District in the Czech Republic [140], and in the Sulcis and Arburese areas of Sardinia [141–143]. All these ore deposits are related to post-Variscan and late Carboniferous to Permian magmatic-hydrothermal events. Post-Variscan mineralization events are also documented in Europe, e.g., the Middle Permian and Mesozoic ones in southern Sardinia [144].

Indeed, a more detailed study could help to constrain the genetic conditions and to highlight any possible correlation among different mineralization sites/events, also with the aim of ascertaining such a largely accepted supposed connection of the Calabrian mineralization with those found in other European regions with similar geological context, such as Sardinia and the French Central Massif (e.g., [56,108,112]. In addition, detailed research of the Calabrian ore deposits might contribute to better understanding the possible association of the Calabria geodynamic context with that of other European Variscan Massifs.

In this respect, the spatial relationship existing between most of the Calabria ore mineralizations and the late Variscan intrusions was observed by several authors (e.g., [53,55]), even though their genetic connection is still to be analytically proved. The hypothesis put forward in past times of a genetic link to porphyry type deposits was discarded based on fluid inclusion studies, at least for the Sila mineralization occurrences [106–108]. In particular, a wide range of temperatures (50–416 °C) and salinities (0–26 wt.% NaCl eq.) were observed for fluid inclusions entrapped within magmatic rocks from different sites of the Sila batholith [107]. It was suggested that the highest temperatures might correspond to magmatic-related Variscan fluids, and the lowest ones to fluids of either Variscan or Alpine age. Fluids typical of porphyry systems were not observed [107]. In spite of this, the possibility that non-porphyry fluids were linked to the Variscan intrusions, because they at least suffered their heating effects, is another genetic hypothesis to be verified.

Moreover, a further possibility is that the ore deposits are chronologically subsequent to the Variscan intrusions. All these hypotheses require verification by means of scientific-analytical studies, considering that one other big open question just deals with the lack in dating studies of the ore mineralization. Furthermore, the conclusions proposed by some authors (e.g., [55]) that primary concentrations of ore minerals genetically linked to Variscan intrusions were subsequently dispersed by the Alpine metamorphism, and that the scarcity of ore mineralization in the Alpine ophiolitic units (e.g., in the Gimigliano Unit) might be partly due to this effect, are still to be verified.

Strictly linked to the genetic topic, scientific studies aimed at unravelling the relationships existing between the vein-type mineral deposits and the tectonic evolution, leading to the formation of the mineralized fractures, are still needed and represent future perspectives.

The mining industry in the Sila and Serre areas has always been poor and concentrated in a few sites, especially in the Stilaro valley and in the Longobucco area, mainly because of the inaccessible nature and transport difficulties of those regions that made the extractive activities too much expensive, and also considering the small ore volumes [8,28,37]. Furthermore, even though the old literature came to the conclusion that the Calabria-Peloritani Arc was to be considered a barren region due to the small size of its mineralizations [56], a renewed economic interest in these areas may arise by considering the possibility that sphalerite occurrence in historical mining districts may represent an attractive target for critical metals such as Ge, Ga and In [143]. Indeed, a wide range of minor/trace elements (including Ga, Ge, In) can substitute for Zn in sphalerite, and, in some cases, these elements are extracted as by-products during sphalerite processing [145]. For example, the sphalerite of the main Ge deposit in western Europe, the vein-type Zn–Ge–Ag–(Pb–Cd) deposit of Noailhac-Saint-Salvy (Tarn, France), contains up to 2600 ppm Ge [135,146].

In our opinion, in spite of the "scarce" economic potential of the Calabria ore deposits, their importance from a scientific view point must still be preserved and appreciated. Scientific analytical data have been published only for a few mineralization sites. In many cases, the available data were collected a long time ago and presented in pioneering papers (e.g., [106]). In the majority of cases, studies were abruptly ceased at the beginning of the 1990′s (e.g., [108]), possibly because of the declared loss of economic interest for extractive activities in these areas. In a few recent cases, renewed scientific interest for the Calabria mineralization led to the publication of short analysis reports (e.g., [117,118]), with a great temporal gap with respect to the older literature. This led to a considerable open space for further analytical research on these mineral deposits, with the aim of a proper characterization, possibly leading to ascertaining genetic hypotheses on their formation.

**Author Contributions:** Conceptualization, V.F. and R.A.F.; investigation, R.A.F., A.C. and F.T.; resources, R.A.F., A.C. and F.T.; writing—original draft preparation, R.A.F. and A.C.; writing—review and editing, R.A.F., A.C., V.F., G.R., F.T., E.S. and G.V.; visualization, A.C., V.F., F.T. and R.A.F.; project administration, R.A.F.; funding acquisition, R.A.F. and A.C. All authors have read and agreed to the published version of the manuscript.

**Funding:** This research was funded by Università degli Studi di Bari Aldo Moro, grant numbers U.P.B. FregolaRicerca15-16, and U.P.B. FregolaRicat17-18 to R.A.F.; Ph.D. grant to A.C.

**Data Availability Statement:** Data are contained within the article.

**Acknowledgments:** We are grateful to three anonymous reviewers, whose comments and suggestions helped us to improve the original draft of the manuscript. The Editors are thanked for the handling of the manuscript.

**Conflicts of Interest:** The authors declare no conflict of interest. The funders had no role in the design of the study; in the collection, analyses, or interpretation of data; in the writing of the manuscript; or in the decision to publish the results.

## Appendix A

**Table A1.** Mineralization sites and geological framework of the Sila Massif.

| Literature Data | | | | | | | | | Review |
|---|---|---|---|---|---|---|---|---|---|
| Mineralization Type (**) | No. | Locality | Ore Deposit | Ore Minerals | Gangue Minerals | Host Rock and Old Tectonic Unit | Assumed Genesis | References | Contemporary Tectonic Unit |
| Pre-Alpine within magmatic rocks | 1 | Torrente Grammisate (Rossano) | Zn-Pb-Fe-Cu | sp, gn, ccp, | flr | veins within Variscan granitoids close to the contact with phyllites; Longobucco Unit | hydrothermal | [39,53,55] | Sila-Serre Unit, Batholith |
| | 2 | Santuario del Patire (Corigliano) | Zn-Pb-Fe (*) | sp, gn, py | qz | | | [54,110] | Sila-Serre Unit, Batholith |
| | 3 | Cerasaro-Cozzo Chinico (Rossano) | Fe | py | qz | | | [39,53–55] | Sila-Serre Unit, Batholith |
| | 4 | Cozzo del Pesco (Corigliano) | Fe-Zn-Pb | py, sp, gn | qz | | | [39,53–55] | Sila-Serre Unit, Batholith and Bocchigliero/ Stilo-Pazzano Unit |
| | 5 | Baracconi—Fonte Argentila—Torrente Petraro (Corigliano) | Pb (*) | gn, (py) | qz | | | [39,53–55,110] | Sila-Serre Unit, Batholith |
| | 6 | Difesella del Trionto (Longobucco) | Zn-Pb | sp, gn | cal | | | [54,55] | Sila-Serre Unit, Batholith |
| | 7 | Croce della Reghinella— Macchiafarna— Fiume Trionto (Longobucco) | Zn-Pb | sp, gn | n.d. | | | [54] | Sila-Serre Unit, Batholith and Bocchigliero/ Stilo-Pazzano Unit |
| | 8a | Torrente La Manna— Torrente Macrocioli— M. Cerzito—Cozzo Vitalba—Fosso Belvedere (Longobucco) | Zn-Fe-Pb-Cu (*) | sp, gn, py, ccp | flr | | | [53,54] | Sila-Serre Unit, Batholith and Bocchigliero/ Stilo-Pazzano Unit |
| | 9 | Serra Toppale (Savelli) | Pb-Zn-Fe (*) | gn, sp, py | n.d. | | | [54] | Bocchigliero/ Stilo-Pazzano Unit |

**Table A1.** *Cont.*

| | | | | | Literature Data | | | Review |
|---|---|---|---|---|---|---|---|---|
| Mineralization Type (**) | No. | Locality | Ore Deposit | Ore Minerals | Gangue Minerals | Host Rock and Old Tectonic Unit | Assumed Genesis | References | Contemporary Tectonic Unit |
| Pre-Alpine within magmatic rocks | 11 | Vallone San Lorenzo (Cerenzia-Castelsilano) | Mo | mol | n.d. | veins within granitoids close to phyllites; Longobucco Unit | hydrothermal | [53–55,115] | Sila-Serre Unit, Batholith |
| | 13 | Cerenzia—Caccuri | Pb-Zn-As-Fe-Cu | gn, sp, py, apy, ccp | qz, flr, cal | | | [29,52–54] | Sila-Serre Unit, Batholith |
| | 12 | San Giovanni in Fiore (CS) | Pb-Fe (U) | gn, py, (maut) | flr | granitic rocks (incrustation) | n.d. | [3,29,114] | Sila-Serre Unit, Batholith |
| | 14 | Molino Mastricarro (Catanzaro) | Ba (*)-Pb-Cu | brt, gn, ccp | qz, flr | veins within granitic and porphyritic rocks; Stilo Unit | epithermal | [15–17,36,54,55,106,116–118] | Sila-Serre Unit, Batholith |
| Pre-Alpine within metamorphic rocks | 8b | Cozzo Vitalba (Longobucco) | Fe | py | n.d. | veins in carbonatic lenses intercaleted to phyllites; Longobucco Unit | n.d. | [53] | Bocchigliero/ Stilo-Pazzano Unit |
| | 8c | Croce Reinella (Longobucco) | W | sch | n.d. | skarns in phyllites; Bocchigliero Unit | n.d. | [107,112,113] | Bocchigliero/ Stilo-Pazzano Unit |
| | 10a | Savelli | Fe | py | n.d. | hornfels; Bocchigliero Unit | contact-metamorphic. | [39,53,55] | Mandatoriccio/ Mammola Unit |
| | 10b | Savelli | W-Zn-Cu | n.d. | n.d. | skarns | contact-metamorphic | [112] | Mandatoriccio/ Mammola Unit |
| | 10c | Torrente Sanapite (Savelli) | Cu-Fe-Pb | ccp, mlc, gn, py | n.d. | pegmatitic dikes in paragneiss; Mandatoriccio Unit | n.d | [53] | Mandatoriccio/ Mammola Unit |
| | 15 | Tiriolo | Fe-Cu-Zn | py, ccp, sp, ttr, bn, mlc, lm | n.d. | meta-carbonate rocks close to the granite intrusion; Stilo Unit | n.d. | [15,17,19,20,24,25,29,53,122,126] | Sila-Serre Unit, Lower Crust |
| | 16 | Colle Stretto, Colle Pallone, Fosso Patia (Gimigliano) | Fe (*) | py | n.d. | phyllites and phyllitic quartzites; Bagni Unit | sedimentary; metamorphic | [2,28,29,53–55,109] | Fiume Pomo Unit |

**Table A1.** *Cont.*

| | | | | | | | | Literature Data | | | | Review |
|---|---|---|---|---|---|---|---|---|---|---|---|---|
| Pre-Alpine within metamorphic rocks | 17 | S. Pietro Apostolo, Martirano, Conflenti | Fe | py | n.d. | metamorphic rocks; Bagni Unit | metamorphic | [28,54,126] | | | | Fiume Pomo Unit |
| | 18 | Platania (CZ) | Fe | py | qz | lenses in quartz-rich phyllites; Bagni Unit | hydrothermal | [29,53,55,130,131] | | | | Fiume Pomo Unit |
| | 19 | Serra Pedace (CS) | Pb, (Ag) | (Ag)-gn | n.d. | phyllites | n.d. | [23] | | | | Sila-Serre Unit, Lower Crust |
| Recent | 20 | V.ne Cupone-Cozzo del Principe-Gallopane (Longobucco) | U | aut, tor | n.d. | lacustrine sediments; Longobucco Unit | sedimentary (secondary) | [54,55] | | | | Sila-Serre Unit (Quaternary sediments) |

Notes: The table includes all the available literature information. The lack of detailed information regards, e.g., the host rock (sometimes metamorphic rocks are generically reported), and the assumed genesis of the mineral deposit. (*) Historically mined; (**) in relation to the Alpine orogenesis and host rock; n.d. = not defined. Mineral symbols: apy (arsenopyrite), aut (autunite), maut (meta-autunite), bn (bornite), brt (baryte), cal (calcite), ccp (chalcopyrite), flr (fluorite), gn (galena), lm (limonite), mlc (malachite), mol (molybdenite), py (pyrite), qz (quartz), sp (sphalerite), sch (scheelite), tor (torbernite), ttr (tetrahedrite).

# Appendix B

**Table A2.** Mineralization sites and geological framework of the Serre Massif.

| | | | | | Literature Data | | | | Review |
|---|---|---|---|---|---|---|---|---|---|
| Mineralization Type (**) | No. | Locality | Ore Deposit | Ore Minerals | Gangue Minerals | Host Rock and Old Tectonic Unit | Assumed Genesis | References | Contemporary Tectonic Unit |
| Pre-Alpine within magmatic rocks | 22b | Cantiere Giogli (Pazzano, Bivongi) | Mo (Cu, Fe, U) (*) | mol, py, ccp, cv, lm, fmyb, U-molybdate | qz | mineralized dikes in quartz-monzonite | pneumatolytic | [26,38,127–129] | Sila-Serre Unit, Batholith |
| | 22c | Argostile (Stilo) | Fe-As (Pb-Zn) | apy, orp (gn, sp) | qz | mineralized dike in quartz-monzonites (granodiorites), close to the contact with schists | pneumatolytic; (hydrothermal) | [38,55] | Sila-Serre Unit, Batholith and Bocchigliero/ Stilo-Pazzano Unit |
| | 23a | Bagni di Guida, Stilaro (Bivongi) | Mo (Cu-Fe) | mol, (ccp) | n.d. | magmatic rocks; Stilo Unit | hydrothermal | [54] | Sila-Serre Unit, Batholith |
| | 25 | San Todaro | Mo (Cu-Fe) (*) | mol, (ccp) | n.d. | quartz-monzonite; Stilo Unit | hydrothermal | [38,54] | Sila-Serre Unit, Batholith |
| | 26a | Calatria | Mo (*) | mol | qz | dikes in quartz-monzonite; Stilo Unit | hydrothermal | [38] | Sila-Serre Unit, Batholith |

**Table A2.** *Cont.*

| Mineralization Type (**) | No. | Locality | Ore Deposit | Ore Minerals | Gangue Minerals | Host Rock and Old Tectonic Unit | Assumed Genesis | References | Contemporary Tectonic Unit |
|---|---|---|---|---|---|---|---|---|---|
| | | | | | | **Literature Data** | | | **Review** |
| Pre-Alpine within metamorphic rocks | 21 | Fiumara Assi (Guardavalle, Stilo) | Fe-Cu-Zn-Pb | ccp, sp, gn | n.d. | impregnation zone in Paleozoic schists | n.d. | [38] | Bocchigliero/ Stilo-Pazzano Unit |
| | 22a | Serravetta, Torrente Pardalà (Bivongi) | Fe-Cu-Zn (*) | py, ccp, sp | cal | impregnation zone in Paleozoic schists, close to the contact with quartz-monzonite | n.d. | [38] | Sila-Serre Unit, Batholith; Bocchigliero/ Stilo-Pazzano Unit |
| | 23b | Bivongi | Fe-Cu | py, po, ccp | qz | lenses in carbonaceous phyllites/disseminations in basic rocks intercalated with phyllites; Stilo Unit | metamorphosed sedimentary/ metamorphosed volcanogenic | [54] | Bocchigliero/ Stilo-Pazzano Unit |
| | 23c | Bivongi | Cu-Fe | ccp | qz | metamorphic rocks, close to the contact with granodiorites; Stilo Unit | hydrothermal | [54] | Sila-Serre Unit, Batholith; Bocchigliero/ Stilo-Pazzano Unit |
| | 24a | Monte Stella (Stilo) | Zn-Pb-Cu-Fe | sp, gn, ccp | qz | mineralized dike in Paleozoic schists | n.d. | [38] | Bocchigliero/ Stilo-Pazzano Unit |
| | 26b | Calatria | Fe-Zn-Cu-Pb | po, mrc, py, sp, ccp, gn | qz, cal | dike in phillites close to the contact with granite | n.d. | [38] | Sila-Serre Unit, Batholith; Bocchigliero/ Stilo-Pazzano Unit |
| | 27 | Monte Granieri (Calatria) | Cu-Fe | ccp | qz | disseminated lenses in carbonaceous phyllites; Stilo Unit | metamorphosed sedimentary | [54] | Bocchigliero/ Stilo-Pazzano Unit |
| | 28 | San Blasio (Caulonia) | As-Cu-Fe | apy, py, ccp | qz | metamorphic rocks, close to the contact with granodiorites; Stilo Unit | hydrothermal | [54] | Mandatoriccio/ Mammola Unit |
| Alpine | 24b | Monte Stella, Monte Gallo (Stilo, Pazzano) | Fe (Cu-Pb-Zn) (*) | lm, gth, hem | n.d. | at the contact between Mesozoic limestones and Paleozoic phyllites; in fractures across Jurassic to Cretaceous limestones; Stilo Unit | sedimentary/ supergene | [28,38,54] | Longobucco/ Stilo cover |

Notes: The table includes all the available literature information. The lack of detailed information regards, e.g., the host rock (sometimes metamorphic rocks are generically reported), and the assumed genesis of the mineral deposit. (*) Historically mined; (**) in relation to the Alpine orogenesis and host rock; n.d. = not defined. Mineral symbols: apy (arsenopyrite), cal (calcite), ccp (chalcopyrite), cv (covellite), fmyb (ferrimolybdite), gn (galena), gth (goethite), hem (hematite), lm (limonite), mol (molybdenite), mrc (marcasite), orp (orpiment), py (pyrite), po (pyrrhotite), qz (quartz), sp (sphalerite).

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
