# Peer review of "Review of Polymetallic Mineralization in the Sila and Serre Massifs (Calabria, Southern Italy)"

_minerals, doi:10.3390/min13030439_

Round 1
Reviewer 1 Report
The submitted manuscript is a highly up-to-date summary and discussion of the mineralization of the crystalline regions of Calabria.
Please correct:- the minor orthographic errors and change some small -see attached file
-problems in the figures 2 and 3:some lines cannot be distinguished
To my opinion, do not use mineralization occurrences-I would prefer "mineralization". However, this is only a suggestion.

Author Response
"Please see the attachment."

Reviewer 2 Report
Review of, Fregola et al:
Review of Polymetallic Mineralization in the Sila and Serre 2 Massifs (Calabria, Southern Italy).
This is an interesting review paper to read and from my point of view it is well written and easy to read. The core of the paper is the overview of the known literature on mineralization occurrences of the Sila and Serre massifs allowed us to evidence some critical aspects, as well as to highlight some open issues which are worthy to be dealt with, to deepen the knowledge on such topic. Some comments to authors are highlighted in yellow as seen manuscript text as attached file.
General Comments:
It would be interesting to see the article contains some recommendations for researchers. To provide cues for future studies and to address the following: origin and age of mineralization, more In-depth Investigation for critical elements hosted by the Sila and Serre mineralization….etc.
In Discussion, line 599-602 that you're referred the literature on this subject is scarce, but References list of the manuscript provides about 144, there are enough references.
The last reference [144] not mentioned in the (body of manuscript).
The paper is well written, presents good data for Polymetallic Mineralization in Southern Italy and can be published with minor revisions.
Good Luck
Reviewer

Author Response
"Please see the attachment"

Reviewer 3 Report
Lines 13-26: Abstract is ok but it is recommended to use “mineralizations” in the plural form. As far as you present a new review of ore materials for use by everyone, you need to phrase your sentences in the present simple tense.
Line 27, Keywords: You need to modify your keywords so they should use "ore genesis" instead of "genesis" alone. Also, you need to include tectonic environments among the selected words because the mineralizations you review belong to a variety of tectonic regimes, not to a single one.
Lines 29-81: Introduction is acceptable and it needs some minor corrections as shown in the attached annotated pdf. Literature review covers the available published works on the two investigated massifs (Sila and Serre). It is recommended here to summarize the different tectonic provinces of mineralizations, and also you need to report the main lithologies for different types of mineralizations with respect to the dominating metals in each type.
Geological setting
Line 90: Please use high pressure-low temperature (HP-LT). When you mention about an expression for the first time in your manuscript, you need to write it in full plus abbreviations between two brackets. After this, you are free to use the abbreviated form all over the text.
Line 94: Are the augen gneisses granitic in composition?. If so you can mention it.
Lines 99-100, Fig. 1: The major map needs latitudes and longitudes similar to its inset map.
Line 109 (Fig. 2) and line 126 (Fig. 3): The maps are ok but lats and longs are missing.
Line 157: You need here to emphasize if they are actually granitoids or proper granites are also included. If so, it is preferable to use the term granitic instead. For instance, line 148 (caption of Fig. 4), and as far as they are peraluminous then they are most probably proper granites. Also, line 162: Please check for nomenclature so that you need to modify in parts and unify as well.
Lines 166-167: Try to avoid using "high-grade" to describe intensity or abundance of shear zones as a tectonic/structural domains. Therefore, it is better to use the adjectives, e.g. intense or frequent or abundant.
Literature review
Line 186: The sub-title can be modified into: Polymetallic mineralizations: a literature review.
Line 261: You need to insert a reference or two here to support the dynamic metamorphism of the felsic injections.
Line 354: Why the mineralized dykes are not considered as quartz veins that traverse the granitic lithologies?.
Line 362: What kind of porphyry do you mean?. Is it quartz-feldspar porphyry or else?.
Line 384 and line 443: Always subscript the equivalent NaCl as eq. and as an italics form, which is the routine followed for salinity reported in any fluid inclusions data.
Line 419: You do not describe textures of either pegmatitic or aplitic here so you can use their name directly as pegmatite and aplite.
Line 421, and other lines: Baryte as a non-metallic ore mineral occurs in the form of veins, so please try to avoid to describe the barite-bearing discordant bodies as dykes. The same in line 531 for the As-Fe-Sb ores hosted by quartz-monzonite.
Line 577: Do you mean a hydrothermal origin in the vicinity of quartz vein(s)?.
Line 594, the sub-title can be simple “Discussion” because you already report in this section about the open issues to be considered concerning the spatial distribution in space and time of your mineralizations, as well as styles and the genetic types. Also, you give suggestions for the future studies and no need to report it in the sub-title.
In the appendix tables, replace assumed genesis by “genetic type”. n.d, not determined is not included here in the footnote of the tables.
You should follow the standards for abbreviated ore minerals so that pyrrhotite is not "pyh" as you mention but "po". You can access the following link to achieve this task:
https://www.unige.ch/sciences/terre/research/Groups/mineral_resources/opaques/ore_abbreviations.php
For a review paper, as well as other research papers, the reference list should be prepared carefully so that punctuation should be applied. In your list, sometimes you abbreviate the same journal in different ways and pay attention if you need to abbreviate or give the journal name in full. I did not highlight all mistakes in the list so please refer to the instruction for authors who publish in Minerals, as well as to a recent paper sample, and prepare a good reference list.

Author Response
"Please see the attachment"
